# Align-IQA: Aligning Image Quality Assessment Models with Diverse Human Preferences via Customizable Guidance

## ABSTRACT

The alignment of the image quality assessment (IQA) model with diverse human preferences remains a challenge, owing to the variability in preferences for different types of visual content, including user-generated and AI-generated content (AIGC), etc. Despite the significant success of existing IQA methods in assessing specific visual content by leveraging knowledge from pre-trained models, the intricate factors impacting final ratings and the specially designed network architecture of these methods result in gaps in their ability to accurately capture human preferences for novel visual content. To address this issue, we propose Align-IQA, a novel framework that aims to generate visual quality scores aligned with diverse human preferences for different types of visual content. Align-IQA contains two key designs: (1) A customizable quality-aware guidance injection module. By injecting specializable quality-aware prior knowledge into general-purpose pre-trained models, the proposed module guides the acquisition of quality-aware features and allows for different adjustments of features to be consistent with diverse human preferences for various types of visual content. (2) A multi-scale feature aggregation module. By simulating the multi-scale mechanism in the human visual system, the proposed module enables the extraction of a more comprehensive representation of quality-aware features from the human perception perspective. Extensive experimental results demonstrate that Align-IQA achieves comparable or better performance than SOTA methods. Notably, Align-IQA outperforms the previous best results on AIGC datasets, achieving PLCC of 0.890 (+3.73%) and 0.924 (+1.99%) on AGIQA-1K and AGIQA-3K. Additionally, Align-IQA reduces training parameters by 72.26% and inference overhead by 78.12% while maintaining SOTA performance.

## CCS CONCEPTS

• **Computing methodologies** → **Computer vision tasks**.

## KEYWORDS

Image quality assessment, AI-generated content, human preferences, customizable guidance

**ACM Reference Format:**
Anonymous Author(s). 2024. Align-IQA: Aligning Image Quality Assessment Models with Diverse Human Preferences via Customizable Guidance. In *Proceedings of the 32nd ACM International Conference on Multimedia (MM'24), October 28-November 1, 2024, Melbourne, Australia.* ACM, New York, NY, USA, 10 pages. https://doi.org/10.1145/nnnnnnn.nnnnnnn

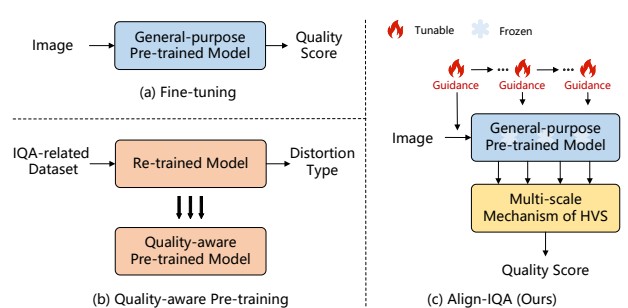

**Figure 1: Comparison of existing "pre-training and fine-tuning" strategies ((a) and (b)) for NR-IQA with our Align-IQA (c).**

## 1 INTRODUCTION

Image quality assessment (IQA) is an indispensable requirement in the fields of image processing and computer vision [35]. It focuses on dealing with different types of visual content, including natural images [30] that have undergone specific treatments such as compression, blurring, noise, *etc.*, user-generated content (UGC) (*e.g.*, images captured through smartphones) [20], and the more recent popularity of AI-generated content (AIGC) (*e.g.*, images produced using text-to-image models) [16]. Over the past few decades, considerable efforts have been invested in proposing various IQA methods. Depending on the necessity of reference images, they can be broadly classified into full-reference IQA (FR-IQA)[25], reduced-reference IQA (RR-IQA)[48], and no-reference IQA (NR-IQA)[19]. In real-world scenarios where reference images are unavailable, NR-IQA is preferred.

In recent years, deep learning-based NR-IQA methods [2, 38, 45] have demonstrated superior performance in evaluating authentically distorted images compared to traditional methods, such as NR-IQA based on natural scene statistics (NSS) [24, 34]. However, their generalization abilities are constrained by the limited size of existing IQA datasets. For instance, the largest authentic IQA dataset, FLIVE[41], encompasses approximately 40,000 distorted images sourced from real-world scenarios. In contrast, the largest image recognition dataset, ImageNet [4], contains more than 14 million labeled images. Accordingly, existing IQA datasets are too small to reflect the diversity and complexity of real-world distortions adequately.

Researchers propose a range of "pre-training and fine-tuning" strategies to address the problem of limited IQA dataset size. As described in Fig. 1 (a), a straightforward way [1, 32] is to directly fine-tune models that are pre-trained on large-scale non-IQA datasets

(*e.g.*, ImageNet) for leveraging knowledge from other computer vision tasks (*e.g.*, image recognition). Nonetheless, it should be noted that these pre-trained models are not specifically designed for IQA, and they focus more on learning semantic-aware representations rather than quality-aware representations. As depicted in Fig. 1 (b), another popular way [22, 29, 47] is to perform quality-aware pre-training, by generating numerous distorted images to simulate real-world degradation processes. Subsequently, a model is pre-trained to identify the distortion type (and level) of these images using self-supervised learning. Since the pre-trained models have already acquired robust quality-aware representations, they could be easily transferred to the downstream NR-IQA task. However, training from scratch requires a lot of training time and computing resources. For instance, a quality-aware pre-training approach [47] based on MoCo-v2[7], takes 75 hours to train ResNet-50 [8] on ImageNet (regenerated by adding synthetic distortions to the image), using 8 Nvidia V100 GPUs.

In summary, we identify three challenges. Firstly, the limited size of the IQA dataset presents difficulties in acquiring a substantial amount of training data. Secondly, there is a struggle to make the pre-trained model pay more attention to extracting quality-aware features rather than semantic-aware features while maintaining computational efficiency. Thirdly, due to the variability in human preferences, it is difficult for existing IQA methods to generate quality scores consistent with diverse human preferences for different types of visual content.

To solve these challenges, we propose Align-IQA, a novel NR-IQA framework aimed at generating visual quality scores aligned with diverse human preferences for various types of visual content. First, we design a customizable quality-aware guidance injection module that integrates specializable quality-aware prior knowledge into general-purpose pre-trained models (*e.g.*, vanilla ViT), where the model is frozen and only a few parameters are introduced for injection. For natural and UGC images, the characteristics of human visual system (HVS) (*e.g.*, visual saliency [33]) are utilized to guide the pre-trained models in learning quality-aware features. In the case of AIGC images, guidance is provided by the visual-semantic relation information obtained from vision-language models. Second, since there are numerous visual natures to affect human perception of image quality [12], we further design a multi-scale feature aggregation module. By simulating the multi-scale mechanism in the HVS, the module enables the extraction of a more comprehensive representation of quality-aware features from the human perception perspective. Specifically, we combine depth-wise separable and dilated convolution, to implement multi-scale feature extraction with a small number of model parameters. Finally, we conduct extensive evaluation and analysis to verify the efficiency and effectiveness of the proposed two designs in Align-IQA. The experimental results show that Align-IQA achieves comparable or better performance than SOTA methods. The main contributions of Align-IQA are summarized as follows:

- We explore a novel NR-IQA model for generating visual quality scores aligned with diverse human preferences for different types of visual content. By using the proposed customizable quality-aware guidance injection module, our method ensures integrating personalized quality-aware prior knowledge into pre-trained models in the frozen state to obtain various quality-aware features for different types of visual content.
- We design a novel multi-scale feature aggregation module. By simulating the multi-scale mechanism in the HVS, the module enables the extraction of a more comprehensive representation of quality-aware features from the human perception perspective. Furthermore, by leveraging depth-wise dilated separable convolution, the module allows multi-scale features to be obtained with only a small number of model parameters.
- Extensive experimental results demonstrate that Align-IQA achieves comparable or better performance than SOTA methods. Notably, Align-IQA outperforms the previous best results on AIGC datasets, achieving PLCC of 0.890 (+3.73%) and 0.924 (+1.99%) on AGIQA-1K and AGIQA-3K. Furthermore, Align-IQA reduces training parameters by 72.26% and inference overhead by 78.12% while maintaining SOTA performance.

## 2 RELATED WORK

The focus of IQA is on extracting quality-aware features aligned with human preference. Before the rise of deep learning, hand-crafted feature engineering dominated the field of NR-IQA. The most common approach for manually extracting quality-aware features is to build NSS models. The assumption underlying this is that the feature distribution of undistorted natural scene images adheres to certain statistical regularities, which are corrupted by various distortions [28]. Therefore, the quality of distorted images can be quantified by modeling distortion-sensitive statistics of natural scenes, including discrete wavelet transform coefficients [34], locally normalized luminance coefficients [23], and correlation coefficients of subbands [24], etc. Besides, some methods [37, 40] have also made beginning efforts to explore the automatic extraction of quality-aware features from distorted images, typically relying on visual codebooks.

Recently, a variety of deep learning-based methods have been further developed to better automatically extract quality-aware features, significantly improving performance in evaluating real-world distortions. In the pioneering work [11], a shallow network of only one convolutional layer is used for NR-IQA. Naturally, later works transition the network architecture from shallow to deep [2, 21]. Because of its non-local self-attention mechanism, ViT was recently adopted to design the NR-IQA model [12, 39]. Concurrently, the CLIP-based method [45] is also sprouting up, based on a language and vision model [27] can exploit the label semantic.

In addition to well-designed models for extracting quality-aware features, a few studies pay special attention to solving the main challenge of deep learning-based NR-IQA methods: the limited size of existing IQA datasets. Among them, some [22, 29, 47] attempt to perform quality-aware pre-training via self-supervised learning (e.g., contrastive learning [3]), based on the large-scale IQA-related dataset created by synthesizing the distorted image. Meanwhile, others [1, 32] resort to directly fine-tuning the model pre-trained

Figure 2: The overall architecture of Align-IQA. First, the input image is passed through the patch embedding layer to obtain image embedding. Among these, image embedding is fed into $L$ transformer layers. Then, customizable quality-aware prior knowledge is mapped to a feature embedding of $D$ dimensions through an additional patch embedding layer, in conjunction with image embedding to the customizable guidance injector (CGI) for generating and injecting the specializable guidance into the transformer blocks for guiding the ViT to focus more on extract quality-aware features rather than semantic-aware features. The multi-scale feature aggregator (MSFA) extracts and integrates multi-scale features from the multi-level outputs of ViT. Finally, the global-local quality-aware features are sent to the dual-branch predictor for final quality prediction.

on large-scale non-IQA datasets, aiming to leverage the knowledge from other computer vision tasks.

We summarize the differences between Align-IQA and the above-mentioned methods. First, different from contrastive learning-based methods [22, 29, 47], Align-IQA does not require re-training from scratch on plenty of IQA-related images, thus avoiding significant computational overhead for re-training. Second, in contrast to methods [1, 32], Align-IQA allows the pre-trained model to prioritize the acquisition of quality-aware features over semantic-aware features, boosting the ability of the model to learn quality-aware representations. Furthermore, by introducing customizable guidance, Align-IQA generates visual quality scores consistent with diverse human preferences for different types of visual content.

## 3 PROPOSED METHOD

### 3.1 Overall Architecture

The proposed Align-IQA is designed to generate visual quality scores aligned with diverse human preferences for different types of visual content. Specifically, for natural and UGC images, the characteristics of HVS (*e.g.*, visual saliency [33]) are utilized to guide the pre-trained models in learning quality-aware features. In the case of AIGC images, guidance is provided by the visual-semantic relation information obtained from vision-language models (*e.g.*, [36]). Align-IQA maintains the transformer encoder of general-purpose pre-trained ViT in a frozen state and contains only a limited set of tunable parameters to learn quality-aware features from the frozen image embedding, guided by customizable quality-aware prior knowledge. The overall architecture of our Align-IQA is illustrated in Fig. 2.

From a hierarchical perspective, we propose the framework of ***encoder - [customizable guidance injector] - [multi-scale feature aggregator] - dual-branch predictor***. The encoder is a vanilla ViT, which consists of a patch embedding layer followed by $L$ transformer layers (see Fig. 2(a)). We design a novel customizable guidance injector, as depicted in Fig. 2(b), to inject quality-aware prior knowledge into ViT. Additionally, we propose a novel multi-scale feature aggregator, shown in Fig. 2(c), which extracts multi-scale features from the multi-level outputs of ViT and fuses them to generate global-local quality-aware features. Finally, the global-local quality-aware features are sent to the dual-branch predictor for obtaining the final quality score.

### 3.2 Customizable Guidance Injector

Due to the variability in human preferences for different types of visual content, we propose the customizable guidance injector (CGI). The main goal of CGI is to incorporate specializable quality-aware prior knowledge into the input of transformer layers, thereby preserving the original architecture of ViT. Notably, through the implementation of CGI, Align-IQA can effectively guide the general-purpose pre-trained ViT to focus more on extracting quality-aware features rather than semantic-aware features. As depicted in Fig. 2, the proposed CGI is integrated into each layer of the backbone network. Formally, providing token sequences $\{E_{img}^l, E_{prior}\}$ consist of the frozen image embedding $E_{img}^l$ and tunable feature embedding $E_{prior}$ mapped by quality-aware prior knowledge. The designed CGI to generate guidance with these token sequences, the process

can be written as:

$$P^l = B^l \left( E^l_{img}, E_{prior} \right), l = 1, 2, \ldots, L \qquad (1)$$

where $B^l$ denotes the $l$-th CGI block. In this way, CGI makes full use of quality-aware prior knowledge to generate effective guidance.

The detailed design of CGI is depicted in Fig. 2 (b), our CGI has two input branches for introducing the token sequences of the frozen image embedding $E^l_{img}$ and tunable feature embedding $E_{prior}$, respectively. Specifically, for the $i$-th transformer layer of ViT, we take the tunable feature embedding $E_{prior}$ as the query, and the frozen image embedding $E^l_{img}$ as the key and value. We use multi-head self-attention to incorporate tunable feature embedding $E_{prior}$ into the frozen image embedding $E^l_{img}$, which can be written as:

$$E_{jq} = W_{jq} \left( norm \left( E_{prior} \right) \right) \qquad (2)$$

$$E_{jk}, E_{jv} = W_{jk} \left( norm \left( E^l_{img} \right) \right), W_{jv} \left( norm \left( E^l_{img} \right) \right) \qquad (3)$$

$$Attention(E_{jq}, E_{jk}, E_{jv}) = softmax \left( \frac{E_{jq}^T E_{jk}}{\sqrt{d_k}} \right) E_{jv} \qquad (4)$$

$$P^l = Concat \left( head_1, head_2, \ldots, head_h \right) W^o \qquad (5)$$

$$\hat{E}^l_{img} = E^l_{img} + \gamma_i P^l \qquad (6)$$

where $norm \left( \cdot \right)$ is LayerNorm, $W_{jq}, W_{jk}, W_{jv}, W^o$ are projection functions, $h = 8$, $head_j = Attention(E_{jq}, E_{jk}, E_{jv})$, $d_k$ is the dimension of the key. Furthermore, we apply a learnable parameter $\gamma_i$ to balance the tunable guidance embedding $P^l$ and the frozen image embedding $E^l_{img}$, which is initialized as $7.68 \times 10^{-3}$. As a result, the feature distribution of $E^l_{img}$ is not significantly altered because of the injection of quality-aware prior knowledge, thereby optimizing the utilization of the pre-trained weights of ViT.

## 3.3 Multi-Scale Feature Aggregator

In the human visual system, there are numerous visual natures to affect human perception of image quality. To better predict quality scores from the human perception perspective, we propose the multi-scale feature aggregator (MSFA). In particular, ViT is divided uniformly into $N$ (usually $N = 4$) blocks, each block containing $L/N$ transformer layers. After injecting quality-aware prior knowledge into ViT, we obtain four output features, $E^3_{img}, E^6_{img}, E^9_{img}, E^{12}_{img}$, by processing the frozen image embedding $E^l_{img}$ through the transformer layers of blocks 1, 2, 3, and 4. Then, we apply a module that combines depth-wise separable convolution and dilated convolution to extract multi-scale features. In this way, by replacing the standard convolution with depth-wise dilated separable convolutions, our MSFA significantly reduces the number of parameters and computational costs. As shown in Fig. 2 (c), the proposed MSFA consists of three main components: (i) depth-wise dilated separable convolution operations that capture multi-scale features; (ii) a cross-level aggregation operation for synthesizing cross-level features; (iii) a $1 \times 1$ convolution operation for the fusion of multi-scale features and cross-level features.

Concretely, there is a $1 \times 1$ standard convolution and three $3 \times 3$ depth-wise dilated separable convolutions, which capture local features of different receptive fields by setting different dilation rates ($r^3 = 3, r^6 = 2, r^9 = 1$, respectively). By introducing spatial inductive bias towards the enrichment of local information. Multi-scale features can be obtained using the following equation:

$$F^j_{img} = \begin{cases} Conv^{r^j}_{dw} \left( Conv_{pw} \left( E^j_{img} \right) \right), j = 3, 6, 9 \\ Conv_{pw} \left( E^j_{img} \right), j = 12 \end{cases} \qquad (7)$$

where $Conv_{pw}$ indicates a $1 \times 1$ point-wise convolution for dimensionality reduction, $Conv^r_{dw}$ indicates a $3 \times 3$ depth-wise dilated convolution for capturing multi-scale feature, $r$ indexes the dilation rate.

Furthermore, we utilize a cross-level aggregation operation to fuse cross-level features across multiple levels of ViT. This process is described as follows:

$$Concat \left[ E^3_{img}, E^6_{img}, E^9_{img}, E^{12}_{img} \right] \longrightarrow F^{cross}_{img} \qquad (8)$$

$$\hat{F}^{cross}_{img} = Upsample \left( Conv^R_{1 \times 1} \left( AvgPool \left( F^{cross}_{img} \right) \right) \right) \qquad (9)$$

where $AvgPool \left( \cdot \right)$ is global average pooling, $Conv^R_{1 \times 1} \left( \cdot \right)$ indicates a $1 \times 1$ convolution with ReLU activation function, $Upsample \left( \cdot \right)$ indexes the bilinear interpolation.

After that, we apply a convolution to coordinate multi-scale branches and the cross-level branch obtaining the integrated global-local quality-aware feature $F_{img}$. The process is formulated as:

$$Concat \left[ F^3_{img}, F^6_{img}, F^9_{img}, F^{12}_{img}, \hat{F}^{cross}_{img} \right] \longrightarrow F_{img} \qquad (10)$$

$$\hat{F}_{img} = Conv_{1 \times 1} \left( F_{img} \right) \qquad (11)$$

where $Conv_{1 \times 1} \left( \cdot \right)$ indicates a $1 \times 1$ convolution function to fuse the multi-scale features and cross-level features.

## 3.4 Prediction

When assessing the quality of images, human observers tend to focus on conspicuous regions of the images, $e.g.$, sharp edges. As a result, different image patches usually have diverse quality scores. To tackle the problem of different image patches exerting varying influences on the final quality score of the entire image, Align-IQA adopts the dual-branch structure of patch-weighted quality prediction, as delineated in AHIQ [14]. We obtain the final predicted quality score by the following equation:

$$\hat{S} = \frac{\sum_{i=1}^N w_i s_i}{\sum_{i=1}^N w_i} \qquad (12)$$

where $N$ denotes the number of patches for a single image, $s_i$ denotes the quality score of the $i$-th patch, $w_i$ denotes the corresponding weight.

## 3.5 Interpretation of Align-IQA

As shown in Fig. 3, we visualize the feature maps of vanilla ViT and the proposed Align-IQA. We perform visualization analyses on a fast-fading distorted image and a JPEG compression distorted image, both of which are derived from the LIVE dataset. Compare

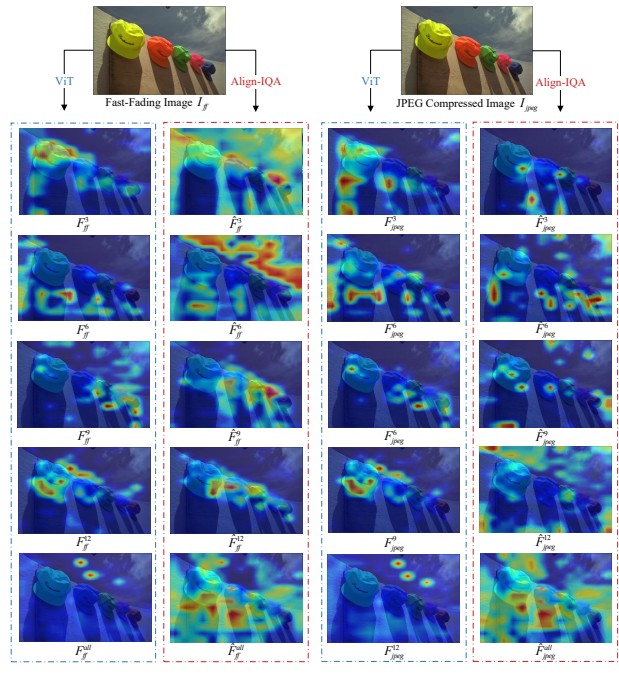

**Figure 3: Visualization of the feature maps from different stages in vanilla ViT and Align-IQA, respectively.**

the outputs from each block in ViT with the corresponding outputs from each block of Align-IQA, the feature maps $F_{ff}^3/F_{jpeg}^3$ to $F_{ff}^{12}/F_{jpeg}^{12}$ increasingly concentrate on the target instance and exhibit enhanced visual-linguistic alignment within the semantic space, while the feature maps $\hat{F}_{ff}^3/\hat{F}_{jpeg}^3$ to $\hat{F}_{ff}^{12}/\hat{F}_{jpeg}^{12}$ do not demonstrate an increasing focus on semantic information, instead, they exhibit a balanced focus on both semantic elements and peripheral details such as edges. It indicates that CGI effectively injects quality-aware prior knowledge into ViT, thereby guiding it to prioritize the acquisition of quality-aware features over semantic-aware features. Impressively, unlike $F_{ff}^{all}$ and $F_{jpeg}^{all}$, in $\hat{F}_{ff}^{all}$ and $\hat{F}_{jpeg}^{all}$, regions of interest are more diffuse and broad, extending far beyond the semantic space itself, which indicates that MSFA can effectively capture global and local information, thereby enriching the representation of quality-aware features from the human perception perspective.

# 4 EXPERIMENTS

## 4.1 Experimental Setups

*4.1.1 Datasets.* We evaluate the performance of Align-IQA on eight publicly available IQA datasets, including synthetic datasets such as LIVE [30], CSIQ [15], TID2013 [26], and KADID-10K [18], authentic datasets like CLIVE [6] and KonIQ-10K [9], as well as AIGC datasets like AGIQA-1K [46] and AGIQA-3K [16]. LIVE consists of 779 distorted images generated from 29 reference images using five distortion types. CSIQ contains 866 distorted images obtained from 30 reference images. TID2013 consists of 3,000 distorted images sourced from 25 reference images, covering 24 distortion

types. KADID-10K contains 10125 images synthetically distorted using 25 different distortions on 81 reference images. CLIVE comprises 1,162 real-world distorted images captured by various mobile devices. KonIQ-10K contains 10,073 images with diverse authentic distortions. AGIQA-1K consists of 1,080 images generated by two text-to-image (T2I) models. AGIQA-3K contains 2,982 AI-generated images produced by six T2I models. For each IQA dataset, 80% of the distorted images contained in it are randomly selected for training and the rest 20% are used to test.

*4.1.2 Metrics.* Spearman's Rank-Order Correlation Coefficient (SR-CC) and Pearson's Linear Correlation Coefficient (PLCC) are employed as metrics to evaluate the performance of Align-IQA. They measure prediction monotonicity and precision, respectively. For both two metrics, their values are in the range of [0,1]. When the value of SRCC or PLCC is closer to 1, it indicates a stronger positive correlation between the predicted quality score of an IQA method and the ground-truth quality score.

*4.1.3 Implementations.* During the training phase, our proposed model Align-IQA is trained for 300 epochs with a batch size of 4. The adamW optimizer with a weight decay of $10^{-5}$ is employed. The learning rate is initialized with $10^{-4}$ and scheduled by the cosine annealing strategy. Since we use ViT-Base/16 [5] model pre-trained on ImageNet as the backbone network of Align-IQA, we randomly crop all input images into three sub-images with a size of $224 \times 224$. The training loss is computed by the mean square error loss. During the test phase, we randomly crop each image 20 times, and the final quality score is calculated as the mean of the quality scores from each cropped sub-image. The above implementations are all completed using Python and PyTorch on 2 Nvidia A100 GPUs.

## 4.2 Comparisons with the SOTAs

For synthetic and authentic datasets, we compare the proposed method with six fully fine-tuned NR-IQA models, including LIQE [45], MANIQA [39], MUSIQ [12], UNIQUE [44], KonCept[9], and HyperIQA[31], two quality-aware pre-trained models, including Re-IQA [29], and CONTRIQUE [22], two well-designed CNN-based models, including NSSADNN [38], and WaDIQaM-NR [2], as well as two hand-crafted feature-based models, including BIECON [13], and HOSA [37]. We report the SRCC and PLCC results in Tab. 1. For AIGC datasets, the comparison results are reported in Tab. 2 and Tab. 3. From the results, we draw some conclusions. First, the quality-aware pre-training strategy enables CONTRIQUE and Re-IQA to outperform well-designed CNN-based WaDIQaM-NR, as well as UNIQUE based on a general-purpose pre-trained model. Second, by introducing textual modal information, LIQE performs much better than CONTRIQUE and Re-IQA on CLIVE and KonIQ-10K datasets. Furthermore, Align-IQA outperforms CONTRIQUE, Re-IQA, LIQE, transformer-based MANIQA, and MUSIQ, as well as AIGC-related PSCR[42] and TIER[43] on the various datasets, which verifies the effectiveness of our Align-IQA, injecting quality-aware prior knowledge into the general-purposed pre-trained model and extracting multi-scale quality-aware features. Notably, our Align-IQA uses significantly fewer training parameters (35.43 M) than the latest quality-aware pre-trained models and fully fine-tuned

**Table 1: Performance comparisons between the proposed Align-IQA and existing SOTA NR-IQA methods on the LIVE, CSIQ, TID2013, KADID-10K, CLIVE, and KonIQ-10K datasets. The best and the second-best performance results are marked in boldface.**

| Method | Synthetic Distortions | | | | | | | | Authentic Distortions | | | |
| | LIVE | | CSIQ | | TID2013 | | KADID-10K | | CLIVE | | KonIQ-10K | |
| | SRCC | PLCC | SRCC | PLCC | SRCC | PLCC | SRCC | PLCC | SRCC | PLCC | SRCC | PLCC |
|---|---|---|---|---|---|---|---|---|---|---|---|---|
| HOSA[37] | 0.946 | 0.947 | 0.741 | 0.823 | 0.735 | 0.815 | 0.618 | 0.653 | 0.640 | 0.678 | 0.780 | 0.795 |
| BIECON[13] [13] | 0.961 | 0.962 | 0.815 | 0.823 | 0.717 | 0.762 | - | - | 0.595 | 0.613 | - | - |
| WaDIQaM-NR[2] | 0.954 | 0.963 | - | - | 0.761 | 0.787 | - | - | 0.671 | 0.680 | 0.739 | 0.761 |
| NSSADNN [38] | **0.984** | **0.986** | 0.927 | 0.893 | 0.910 | 0.844 | - | - | 0.813 | 0.745 | - | - |
| CONTRIQUE[22] | 0.960 | 0.961 | 0.942 | 0.955 | 0.843 | 0.857 | **0.934** | **0.937** | 0.845 | 0.857 | 0.894 | 0.906 |
| Re-IQA[29] | 0.970 | 0.971 | 0.947 | 0.960 | 0.804 | 0.861 | 0.872 | 0.885 | 0.840 | 0.854 | 0.914 | 0.923 |
| HyperIQA[31] | 0.962 | 0.966 | 0.923 | 0.942 | 0.840 | 0.858 | 0.852 | 0.845 | 0.859 | 0.882 | 0.906 | 0.917 |
| KonCept[9] | 0.673 | 0.619 | 0.631 | 0.645 | - | - | 0.503 | 0.515 | 0.778 | 0.799 | 0.911 | 0.924 |
| UNIQUE[44] | 0.961 | 0.952 | 0.902 | 0.921 | - | - | 0.884 | 0.885 | 0.854 | 0.884 | 0.895 | 0.900 |
| MUSIQ [12] | 0.837 | 0.818 | 0.697 | 0.766 | - | - | 0.572 | 0.584 | 0.785 | 0.828 | 0.915 | **0.937** |
| MANIQA[39] | 0.982 | 0.983 | **0.961** | **0.968** | **0.937** | **0.943** | **0.944** | **0.946** | 0.890 | **0.910** | **0.920** | **0.943** |
| LIQE[45] | 0.970 | 0.951 | 0.936 | 0.939 | - | - | 0.930 | 0.931 | **0.904** | **0.910** | 0.919 | 0.908 |
| Align-IQA | **0.985** | **0.987** | **0.975** | **0.981** | **0.955** | **0.960** | 0.928 | 0.932 | **0.905** | **0.916** | **0.923** | 0.932 |

**Table 2: Performance comparisons on the AGIQA-1K dataset. The best and the second-best performance results are marked in boldface.**

| Method | AGIQA-1K | |
| | SRCC | PLCC |
|---|---|---|
| ResNet50[46] | 0.637 | 0.732 |
| StairIQA[46] | 0.550 | 0.609 |
| MGQA[46] | 0.601 | 0.676 |
| WaDIQaM-NR[2] | 0.728 | 0.779 |
| CONTRIQUE[22] | 0.793 | **0.858** |
| PSCR[42] | **0.843** | 0.840 |
| TIER[43] | 0.827 | 0.830 |
| Align-IQA | **0.855** | **0.890** |

**Table 3: Performance comparisons on the AGIQA-3K dataset. The best and the second-best performance results are marked in boldface.**

| Method | AGIQA-3K | |
| | SRCC | PLCC |
|---|---|---|
| DBCNN[16] | 0.821 | 0.876 |
| CLIPIQA[16] | 0.843 | 0.805 |
| CNNIQA[16] | 0.748 | 0.847 |
| WaDIQaM-NR[2] | 0.219 | 0.393 |
| CONTRIQUE[22] | 0.807 | 0.887 |
| PSCR[42] | **0.850** | **0.906** |
| TIER[43] | 0.825 | 0.882 |
| Align-IQA | **0.874** | **0.924** |

models, while achieving higher PLCC value on the CSIQ dataset, as shown in Fig. 4. Additionally, Align-IQA outperforms the previous best results on AIGC datasets, achieving PLCC of 0.890 (+3.73%) and 0.924 (+1.99%) on AGIQA-1K and AGIQA-3K.

## 4.3 Cross-Dataset Validation

To demonstrate the generalization ability of our Align-IQA, we conduct cross-dataset tests. Specifically, HOSA [37] and WaDIQaM-NR [2] are selected for comparison, with the SRCC results presented in Tab. 4. Among the synthetic cross-dataset validations, Align-IQA achieves the highest performance. With the help of quality-aware prior knowledge and multi-scale mechanism in the HVS, Align-IQA can accurately evaluate the quality of images with unseen distortion types. Cross-dataset validation shows that our Align-IQA effectively alleviates the problem of limited IQA dataset size.

**Table 4: Comparison of SRCC results on cross-dataset validations.**

| Train | Test | HOSA | WaDIQaM-NR | Align-IQA |
|---|---|---|---|---|
| LIVE | CSIQ | 0.596 | 0.704 | **0.813** |
| | TID2013 | 0.470 | 0.462 | **0.612** |
| CSIQ | LIVE | 0.786 | - | **0.938** |
| | TID2013 | 0.341 | - | **0.582** |
| TID2013 | LIVE | 0.844 | 0.817 | **0.908** |
| | CSIQ | 0.609 | 0.690 | **0.703** |

## 4.4 Ablation Studies

*4.4.1 The effectiveness of each component of Align-IQA.* As reflected in Tab. 5, we undertake individual experiments to analyze the effectiveness of each component of our proposed method.

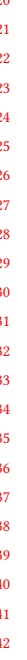

Figure 4: Comparison of our Align-IQA with LIQE, Re-IQA, and MANIQA on Params, GFLOPs ( obtained by the python module "thop"), and PLCC.

The results demonstrate that our CGI module effectively incorporates prior knowledge into the general-purpose pre-trained model, thereby guiding it to focus more on extracting quality-aware features rather than semantic-aware features. Moreover, our MSFA model successfully enriches quality-aware features from the human perception perspective, highlighting the significance of integrating multi-scale information for accurate quality assessment. By simulating the multi-scale mechanism in the HVS, our Align-IQA achieves a further improvement in accuracy.

*4.4.2 The effectiveness of different backbone networks.* We conduct comparative experiments with three types of backbone networks and the results are provided in Tab. 6. The backbone networks used for experiments include ViT-Tiny/16, ViT-Small/16, and ViT-Base/16[5]. For all backbone networks, the input image size is set to 224 × 224, and the shape of the image patch is defined as 16 × 16. It can be found that using ViT-Base/16 works best. This means that using a deeper and wider backbone provides more space to capture richer quality-aware representations, thereby improving performance.

*4.4.3 The effectiveness of different injection strategies.* To evaluate the effectiveness of our CGI module, two additional injection strategies [10] (Fig. 5 (a) & (b)) are selected to introduce guidance tokens into the general-purpose pre-trained model. One strategy involves the summation of the input guide and image embedding. The other strategy entails concatenating the input guide and image embedding. It should be noted that the input guide is replaced with the embedding of visual saliency. The performance comparison between these two injection strategies and the CGI module is depicted in Tab. 7. It is observed that our CGI module achieves the best performance.

*4.4.4 The effectiveness of different strategies for extracting multi-scale features.* To test the effectiveness of our MSFA in building a feature pyramid using the multi-level output of ViT, an additional strategy [17] (refer to Fig. 7 (a)) is selected for comparison. In this strategy, only the output from the last layer of ViT is utilized to build the feature pyramid. The SRCC and PLCC results are summarized in Table 8. It is observed that enhanced performance is achieved

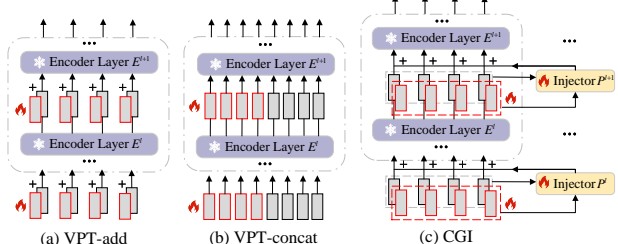

(a) VPT-add    (b) VPT-concat    (c) CGI

Figure 5: Variants of vanilla injection-structure and our CGI.

Table 5: Comparison of each component of our Align-IQA on the KADID-10K dataset.

| CGI | MSFA | KADID-10K | |
|---|---|---|---|
| | | SRCC | PLCC |
| × | × | 0.907 | 0.909 |
| ✓ | × | 0.914 | 0.918 |
| ✓ | ✓ | **0.928** | **0.932** |

Table 6: Comparison of different backbone networks employed by our Align-IQA on CLIVE and KADID-10K datasets.

| | CLIVE | | KADID-10K | |
|---|---|---|---|---|
| | SRCC | PLCC | SRCC | PLCC |
| ViT-Tiny/16 | 0.875 | 0.891 | 0.852 | 0.850 |
| ViT-Small/16 | 0.888 | 0.900 | 0.899 | 0.898 |
| ViT-Base/16 | **0.905** | **0.916** | **0.928** | **0.932** |

Table 7: Comparison of different injection strategies employed by our Align-IQA on CLIVE and KADID-10K datasets.

| | CLIVE | | KADID-10K | |
|---|---|---|---|---|
| | SRCC | PLCC | SRCC | PLCC |
| VPT-add | 0.592 | 0.620 | 0.472 | 0.506 |
| VPT-concat | 0.896 | 0.901 | 0.886 | 0.890 |
| CGI | **0.905** | **0.916** | **0.928** | **0.932** |

through the use of multi-level output for the extraction of multi-scale features. Notably, the sum of SRCC and PLCC values increases by 9.46% and 9.04% in the CSIQ and CLIVE datasets, respectively. This indicates that our Align-IQA can assess image quality more robustly, by comprehensively considering the multi-level features of ViT.

## 4.5 Visualization

Fig. 6 depicts the visual results of the feature maps from VPT-add, VPT-concat, vanilla ViT, and our Align-IQA. We perform visualization analyses on a fast-fading distorted image $I_{ff}$ and a JPEG compression distorted image $I_{jpeg}$, both of which are derived from the LIVE dataset. As can be seen, when vanilla ViT handles the

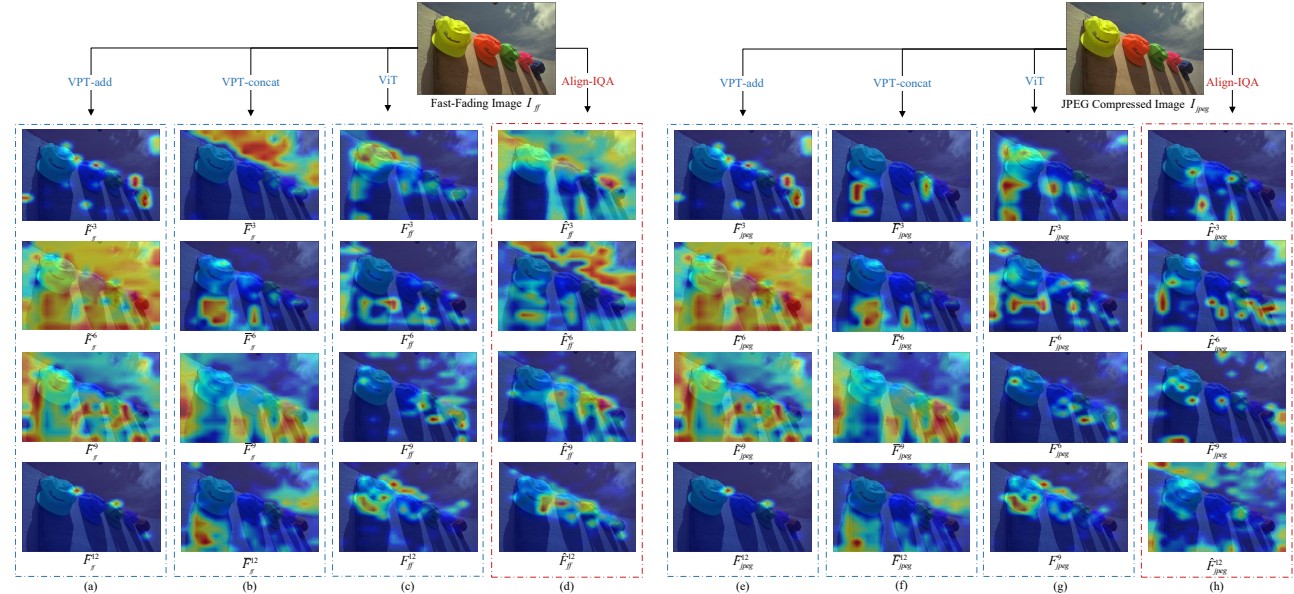

**Figure 6: Visualization of the feature maps from different stages in VPT-add, VPT-concat, vanilla ViT, and Align-IQA, respectively.**

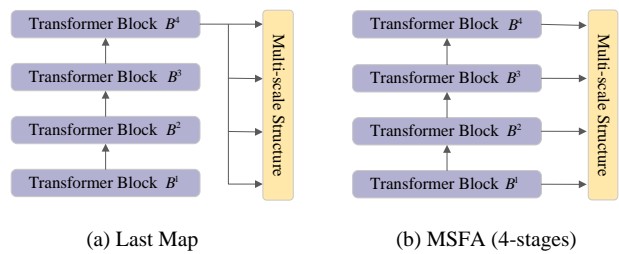

(a) Last Map   (b) MSFA (4-stages)

**Figure 7: Building a feature pyramid on the vanilla ViT. (a) Using solely the final feature map of the vanilla ViT. (b) Our MSFA: the vanilla ViT is artificially segmented into multiple stages.**

**Table 8: Comparison of different strategies for building a feature pyramid on LIVE, CSIQ, TID2013, and CLIVE datasets.**

| Strategy | Last Map | | MSFA(4-stages) | |
|---|---|---|---|---|
| | SRCC | PLCC | SRCC | PLCC |
| LIVE | 0.981 | 0.978 | **0.985** | **0.987** |
| CSIQ | 0.885 | 0.902 | **0.975** | **0.981** |
| TID2013 | 0.911 | 0.922 | **0.955** | **0.960** |
| CLIVE | 0.820 | 0.850 | **0.905** | **0.916** |

fast-fading and JPEG compression distortions, most of the features $F_{ff}^j$ and $F_{jpeg}^j$ extracted at the same stage are similar. The same observation is noted for VPT-add and VPT-concat. For example, the corresponding lines of (a) and (e) are similar respectively, as do the second, third, and fourth lines of (b) and (f). However, the features $\hat{F}_{ff}^j$ and $\hat{F}_{jpeg}^j$ extracted by our Align-IQA at the same stage exhibit

significant variance for different types of distorted images. This is consistent with each distortion type having its own specific way of affecting image quality, indicating that our Align-IQA can extract discriminative features to accurately describe different types of distortion.

## 5 CONCLUSION

In this work, we propose Align-IQA, a novel NR-IQA framework that aims to generate visual quality scores aligned with human preferences for different types of visual content. First of all, we propose a customizable guidance injection module. By injecting specializable quality-aware prior knowledge into general-purpose pre-trained models, the proposed module guides the acquisition of quality-aware features consistent with diverse human preferences for various types of visual content. Second, we propose a multi-scale feature aggregation module. By simulating the multi-scale mechanism in the HVS, the proposed module enables the extraction of a more comprehensive representation of quality-aware features from the human perception perspective. Furthermore, by leveraging depth-wise dilated separable convolution, the proposed module achieves high computation efficiency. Finally, we conduct extensive experiments on eight IQA datasets. The results show that Align-IQA achieves comparable or better performance than SOTA NR-IQA methods. Notably, Align-IQA outperforms the previous best results on AIGC datasets, achieving PLCC of 0.890 (+3.73%) and 0.924 (+1.99%) on AGIQA-1K and AGIQA-3K. Meanwhile, Align-IQA achieves SOTA performance while reducing training parameters by 72.26%. We anticipate that this work motivates more insights on integrating professional prior knowledge into general-purpose pre-trained models, improving the performance and interpretability of downstream IQA tasks.

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
