# OpenReview forum: "Align-IQA: Aligning Image Quality Assessment Models with Diverse Human Preferences via Customizable Guidance"
_acmmm.org/ACMMM/2024/Conference — MM2024 Poster_

### Official Review · Reviewer_pXrk · 2024-05-02

**Rating:** 4
**Confidence:** 3

**Summary:**

This paper proposes a novel framework that aims to generate visual quality scores aligned with diverse human preferences for different types of visual content, which contains two key designs: (1) A customizable quality-aware guidance injection module. (2) A multi-scale feature aggregation module.

**Strengths:**

Innovative method and clear writing.

**Limitations:**

1. Please explain in detail how "the characteristics of HVS" and "visual semantic relationship information" are obtained, respectively. Especially in CGI.
2. The synthetic distortion database should be divided into training and testing sets based on reference images.
3. The prediction performance of the proposed method shown in Table 1 is not very outstanding.
4. The comparison methods in Table 4 are a bit lacking, and it is necessary to add results on the synthetic and AIGC databases.
5. Add the results of testing on an authentically distorted database in Section 4.4.1.
6. If the tunable feature embedding of AIGC images is the same as that of authentically distorted images, how does the proposed method perform in predicting the quality of AIGC images?

**Suitability:**

2

---

### Official Review · Reviewer_T7gJ · 2024-05-20

**Rating:** 3
**Confidence:** 3

**Summary:**

The authors propose a quality assessment method called Align-IQA, which is able to have a good fit with different preferences through customized guidance. The article describes in detail the principles of the proposed method and carries out related experiments, ultimately re-emphasizing the superiority of the algorithm.

**Strengths:**

This paper is able to utilize guidance information to deal with quality assessment of different preferences is a relatively novel idea. In addition, the work has also designed a series of modules that may have some validity. Another advantage of the model is the low inference overhead.

**Limitations:**

Details from the abstract:

1) The direct use of abbreviations is not recommended for both SOTA and PLCC at L31 and L33, although these terms are not unfamiliar to readers with some experience.

Defects:

2) The Customizable Guidance Injector module is not explained clearly enough. How does it primarily relate to Diverse Human Preferences? From Fig 2 I can only see the source of Eimg, the source of Eprior is not explained clearly enough.
3) I think the experimental setup at L550 is unreasonable and the most fatal flaw of this paper. Within my knowledge, all quality assessment algorithms should be given a fixed quality score for the same input. However, the authors have introduced randomness in their reasoning (despite the averaging operation adopted here), which obviously leads to inconsistent assessment results. I suggest that the authors still use a fixed centercrop strategy instead of randomcrop.
4) Based on the experimental results in Table 1, it is felt that the performance improvement of the proposed method is more limited. Combined with the randomness in 3), I have reason to think that these results are better results after randomizing multiple experiments.
5) The results of the ablation experiments in Table 5 show that neither CGI nor MSFA is significant. This contradiction is even more prominent when compared to the results of the backbone experiment in Table 6. In addition, this performance may also be affected by the randomness in 3), so this performance is clearly unconvincing.


Overall, the Achilles' heel of this article is the introduction of randomness mentioned in 3), which compromises the results of many of the later experiments. The suggestions in 1) to 5）are hopefully something that the authors will seriously think about and improve.

**Suitability:**

3

---

### Official Review · Reviewer_SfS1 · 2024-05-24

**Rating:** 4
**Confidence:** 3

**Summary:**

This paper designs a novel Align-IQA framework to perform IQA tasks on different types of visual content.  Contains two key designs: (1) Customizable quality-aware boot injection module.  By injecting specialized quality-aware prior knowledge into a general pre-trained model, the proposed module guides the acquisition of quality-aware features and allows different adjustments of the features to comply with different human preferences for various types of visual content. (2) Multi-scale feature aggregation module.  By simulating the multi-scale mechanisms in the human visual system, the proposed module is able to comprehensively characterize the human visual system.

**Strengths:**

1. The guidance injection module proposed in this article can evaluate different types of content, is highly consistent with the human eye's supervisory experience, and has the potential for application in future IQA tasks.

2. The experimental data set is relatively sufficient, and the comparison methods are complete and advanced.

3. The method proposed in this article is used for both traditional visual content and the emerging AIGC Quality Assessment. It can promote further applications of AIGC and is in line with the purpose of this MM conference.

**Limitations:**

1. The way formulas (2)-(6) are written is a bit confusing. In this part, the author first declares the variables, and then puts all five formulas together, but does not describe the relationship between the variables. Especially in the projection functions section, the block diagram shows that this is an important factor for the effectiveness of Align-IQA, but the author did not explain it. It is recommended to write the formulas one by one.

2. There is room for improvement in the article layout. The authors' typography was not tight enough, resulting in important ablations being placed in the supplementary material. Tables 2-8 can be merged separately to save a certain space.

3. Align-IQA proposed by the author has significant advantages in evaluating AIGC, which is a good point. However, using AGIQA-1K/3K verification is slightly outdated. The development of T2I generative models is extremely rapid, and even the best images in 3K (Jun 2023) are only of average quality compared to the current AIGC. Therefore, the author can verify the performance of Align-IQA on the AIGIQA-20K (Apr 2024) database, thereby better proving its applicability to AIGC.

**Suitability:**

3

---

### Meta-Review · Area_Chair_vhmQ · 2024-07-01

**Recommendation:** Accept (Poster)
**Confidence:** 4

**Metareview:**

This paper proposes a quality assessment method able to fit the differences in preferences among uses assessing images through customized guidance. The article describes the proposed method and carries out related experiments. The three reviewers agree on the potential and relevance of the approach and paper. The reviewers see the potential of the approach and its relevance. However, they rise important challenges. These have to do with the added value of the approach, its novelty and clarifications about the writing. The authors clarified most of the issues, but some of them are still open. This is clearly a borderline paper. I really see the potential of the paper, and the novelty of the approach, while not clearly brought forward, is there. Thus, I am inclined to accept it, given that the authors address all the issues risen by the reviewers.